# Association of *Phytophthora* with Declining Vegetation in an Urban Forest Environment

**DOI:** 10.3390/microorganisms8070973

**Published:** 2020-06-29

**Authors:** Mohammed Y. Khdiar, Paul A. Barber, Giles E. StJ. Hardy, Chris Shaw, Emma J. Steel, Cameron McMains, Treena I. Burgess

**Affiliations:** 1Phytophthora Science and Management, Centre for Climate Impacted Terrestrial Ecosystems, Harry Butler Institute, Murdoch University, Murdoch 6150, Australia; moh_bio84@yahoo.com (M.Y.K.); p.barber@arborcarbon.com.au (P.A.B.); g.hardy@murdoch.edu.au (G.E.S.H.); c.shaw@murdoch.edu.au (C.S.); Emma.Steel@murdoch.edu.au (E.J.S.); Cameron.McMains@murdoch.edu.au (C.M.); 2Biology Department, Education College, Iraqi University, Adhamiya, Baghdad 7366, Iraq; 3Arbor Carbon P/L, ROTA Compound off Discovery Way, Murdoch University, Murdoch 6150, Australia

**Keywords:** metabarcoding, bridgehead effect, biological invasions, remote sensing

## Abstract

Urban forests consist of various environments from intensely managed spaces to conservation areas and are often reservoirs of a diverse range of invasive pathogens due to their introduction through the nursery trade. Pathogens are likely to persist because the urban forest contains a mixture of native and exotic plant species, and the environmental conditions are often less than ideal for the trees. To test the impact of different land management approaches on the *Phytophthora* community, 236 discrete soil and root samples were collected from declining trees in 91 parks and nature reserves in Joondalup, Western Australia (WA). Sampling targeted an extensive variety of declining native trees and shrubs, from families known to be susceptible to *Phytophthora*. A sub-sample was set aside and DNA extracted for metabarcoding using *Phytophthora*-specific primers; the remaining soil and root sample was baited for the isolation of *Phytophthora.* We considered the effect on the *Phytophthora* community of park class and area, soil family, and the change in canopy cover or health as determined through sequential measurements using remote sensing. Of the 236 samples, baiting techniques detected *Phytophthora* species from 24 samples (18 parks), while metabarcoding detected *Phytophthora* from 168 samples (64 parks). Overall, forty-four *Phytophthora* phylotypes were detected. Considering only sampling sites where *Phytophthora* was detected, species richness averaged 5.82 (range 1–21) for samples and 9.23 (range 2–24) for parks. *Phytophthora multivora* was the most frequently found species followed by *P. arenaria*, *P. amnicola* and *P. cinnamomi.* While park area and canopy cover had a significant effect on *Phytophthora* community the R^2^ values were very low, indicating they have had little effect in shaping the community. *Phytophthora cinnamomi* and *P. multivora,* the two most invasive species, often co-occurring (61% of samples); however, the communities with *P. multivora* were more common than those with *P. cinnamomi*, reflecting observations over the past decade of the increasing importance of *P. multivora* as a pathogen in the urban environment.

## 1. Introduction

The urban forest includes trees and shrubs in urban parks, road islands, woodlots, abandoned sites and residential areas [1,2]. Urban trees have been recognized to provide critical ecosystem services to human health and environmental quality, particularly with increasing urbanization [3]. However, urban forests through their mix of native and exotic trees [4], proximity to transport hubs [5] and tree nurseries [6], can provide a pathway for invasive pathogens to establish and move into natural ecosystems [4,7]. To prevent the dissemination of invasive species early detection, accurate identification and the ability to track pathogens back to their potential origin is essential [8].

The genus *Phytophthora* contains invasive pathogenic species responsible for tree diseases worldwide. The spread of *Phytophthora* species is a universal problem for nature preservation due to epidemics such as sudden oak death (USA), ramorum blight (UK), *Phytophthora* dieback (Australia) and *Phytophthora* root rot of Proteaceae (fynbos; South Africa). Within the urban forest, trees are under stress from polluting agents [9], mechanical damage, weeds, excess nutrients, climate change [10], and environmental factors such as waterlogging, salinity, flooding and drought [11,12] therefore increasing their vulnerability to pathogens such as *Phytophthora*.

In recent years, numerous new *Phytophthora* species have been described, many of which include either the type isolate and/or additional isolates from an urban or disturbed ecosystem associated with the original description [13,14,15]. The isolation and identification of *Phytophthora* species can be difficult and time-consuming [16]. However, the use of species-specific primers to amplify target organisms for environmental DNA has made this more straight forward. Metabarcoding has been used for *Phytophthora* in natural ecosystems [17,18,19] and to a limited extent in urban environments [14,16].

Perth, Western Australia (WA) is the most isolated and biodiverse city in the world [20]. Over the past three decades more than 17,000 hectares of vegetation have been cleared across the Perth metropolitan region [21]. Much of the native vegetation remaining in the region is susceptible to *P. cinnamomi*. One of the more recently developed areas north of Perth, the City of Joondalup, has many native Banksia woodland vegetation remnants. Remote sensing is a tool that can be used to monitor tree health in urban forests. It has been used to map *Phytophthora* and the distribution of the diseases they cause in different areas of the world [22,23,24]. Remotely sensed data acquired over four years, enabling changes in tree health to be determined, was utilized for our study. We field-validated the data by examining 236 declining groups of plants across 91 parks in the City’s urban forest. The remote sensing data were also used to determine canopy cover. Rhizosphere soil and roots collected from the declining groups of plants were baited for *Phytophthora* species, and eDNA was extracted from a subsample and subject to metabarcoding using *Phytophthora* specific primers [17]. The *Phytophthora* community of the samples was examined to determine if it was influenced by canopy health, canopy cover, park area, park class or the soil family. Parks within the City tend to be either conservation areas, consisting of native plant species, or managed parks which contain a mix of native and exotic plant species and large areas of turf composed of exotic grass species. We hypothesized that *Phytophthora* communities would differ between conservation parks and those that were mostly turf.

## 2. Materials and Methods

### 2.1. Study Area and Sample Collection

The area investigated was the City of Joondalup, located approximately 26 km north of Perth, WA, covering an area of 98.9 km^2^ (Figure 1) and several different soil complexes. The Spearwood soil is a siliceous yellow sand, weakly acidic with some iron oxide (0% CaCO_3_ to 120 cm depth) [25], the Quindalup soil is a calcareous whitish beach sand (8–36% CaCO_3_), while the Karrakatta soil derived from calcareous beach sand (50–70% CaCO_3_) leached to form secondary calcite layers at depth [26].

For this study, the area was delineated by orthorectified multi-spectral imagery. Between 2012 and 2015, high-resolution digital multi-spectral imagery (DMSI) was captured annually with a fixed-wing aircraft across all urban bushlands in Joondalup, as four narrow spectral bands of data (red, green, blue, and near infrared) at a spatial resolution of 0.5 m pixels. The Red Edge Extrema Index (REEI) was calculated using the ratio of near infrared: red bands, and a difference in pixel values from 2015 to 2012 was calculated by subtracting the 2015 image from the 2012 image [27,28]. An increase in pixel values represent an increase in vegetation condition over this period, while a decrease in pixel values indicates a decline in health. A vegetation feature height model (VFHM) was created by subtracting the digital surface model from the digital terrain model, and a canopy layer was produced by classifying all pixels above 3 m in height within the VFHM.

Field validation was undertaken at 236 sampling sites within 91 parks where soil and root samples were collected to confirm the presence of *Phytophthora* species (Figure 1). These samples were collected during summer and autumn from 2014 to 2016, inclusive. At each of the sampling sites a bulked sample was collected by combining several smaller samples (approximately 150 g) of rhizosphere soil collected within 5 m of the trunks of declining trees. The samples were kept in insulated boxes and transported to the laboratory within 8 h, where they were stored at room temperature (20–22 °C) until processing. Sampling targeted five families known to have genera susceptible to *Phytophthora* (Proteaceae, Myrtaceae, Fabaceae, Casuarinaceae and Asphodelaceae). Samples were collected from an extensive variety of declining native trees and shrubs predominately species of *Acacia, Allocasuarina, Banksia, Corymbia, Eucalyptus, Grevillea, Hibbertia* and *Xanthorrhoea.*

### 2.2. Baiting Technique

For each bulked soil sample (236 in total), 300 g soil with fine root samples were placed in 1 L containers (11.5 × 16.5 × 7.5 mm; GENFAC 111 Plastics P/L). The samples were flooded with distilled water (ratio 3:1 water: soil and roots). Floating baits consisted of young leaves from six plant species (*Quercus ilex*, *Q. suber*, *Pimelea ferruginea*, *Poplar* sp., *Scholtzia involucrata* and *Hedera helix*) and containers were incubated at 20–25 °C for 7 days. Leaves were monitored daily and those baits with lesions were removed, dried on paper towels and 2-mm^2^ lesioned sections were placed on NARH, a *Phytophthora* selective agar medium [29]. Plates were incubated in the dark at room temperature and examined daily for colonies typical of *Phytophthora* species. Any *Phytophthora* growth was sub-cultured onto fresh NARH plates and ultimately transferred onto vegetable juice agar (V8A) plates; 100 mL filtered vegetable juice (Campbells V8 vegetable juice; Campbell Grocery Products Ltd., Norfolk, UK), 900 mL distilled water, 0.1 g CaCO_3_, and 17 g Laboratory-grade agar (Becton, Dickenson and Company, Sparks, MD, USA), adjusted to pH 7. The soil was then allowed to dry for 7 days and baited a second time to increase the chance of isolation success [30].

### 2.3. eDNA Extraction from Fine Roots and Metabarcoding

Prior to baiting, a sub-sample was taken, air-dried, and 60–80 g was ground to a fine powder using the TissueLyser LT (Qiagen, Hilden, Germany). Between each sample, the grinding tubes were cleaned by detergent (Pyroneg), rinsed in 0.4 mM HCl for 5 min, rinsed with water, then sprayed with ethanol 70% and allowed to air dry. Controls were ground with water and all samples were stored at −20 °C. DNA extractions were performed using the Mo-Bio PowerSoil DNA isolation kit (Carlsbad, CA, USA) following the manufacturer’s protocol. Three independent runs were conducted. Amplicon pyrosequencing and clustering were conducted as described previously [16,17]. Briefly, sequences that passed quality checks were imported into Geneious vR9 and sorted into separate files based on their unique barcode. Clustering was performed for each barcode and then the consensus sequences (from both runs were combined and aligned using MAFFT within the Geneious program. Identities were assigned to consensus sequences by conducting an internal blast search firstly against a customized reference database (available upon request from the authors at Phytophthora Science and Management, Murdoch University) and secondly against GenBank. Consensus sequences were then sorted into clades (Appendix A, https://idtools.org/id/phytophthora/molecular.php,) and phylogenetic analyses were conducted for each clade using Geneious tree builder. These final identities were considered phylotypes and represent a new species if they did not match any known species in the phylogenetic analysis. Each *Phytophthora* species was classified as either native or introduced based on patterns of diversity and distribution of representative clades, prior literature, and in some cases comparisons of genetic diversity [17]. Since origin, particularly for microbes, can be difficult to discern and is subject to historical sampling and geographic biases, our designations should be considered as working hypotheses; we have thus used the term ‘putative’ to classify these species.

### 2.4. Data Analysis

Two categorical (park class, soil family) and three continuous variables (park area, canopy cover and canopy health) were examined (Table 1, Appendix A). The *Phytophthora* community was down-sampled to the same number of reads (1000 reads per sample) using the rarefy function in the Vegan package of R [31]. Statistical analyses were performed on the presence/absence data to determine if there was a correlation between the five variables and the *Phytophthora* community at the sites using the Adonis function in the Vegan package for R [31]. Jaccard and Bray–Curtis dissimilarity matrices were created and utilized for presence and abundance analyses, respectively. Only sites (*n* = 168) where *Phytophthora* was recovered were used in the analysis. Each dissimilarity matrix had a permutational multivariate analysis of variance (PERMANOVA) performed with 9999 permutations. The environmental variables (categorical) fit the assumption of homogeneity of multivariate dispersions. The Bray–Curtis *Phytophthora* communities were displayed using unconstrained ordination and non-metric multidimensional scaling (NMDS). Ellipses encompassed 95% of the variation in categorical variables. Continuous variables were displayed on NMDS plots using the envfit function. Species co-occurrence was assessed using the cooccur function with a threshold of >1 in the cooccur R package [32]. Sites with the dominant invasive *Phytophthora* species, *P. cinnamomi* and *P. multivora* were visualized using a Venn diagram constructed in R with the VennDiagram package.

## 3. Results

### 3.1. Isolation and Identification

Only four species of *Phytophthora* (*P. nicotianae*, *P. multivora*, *P. boodjera* and *P. arenaria*) were isolated by baiting from 24 sampling sites (10%) from 18 parks (20%) (Table 2). This result was substantially less than the 44 *Phytophthora* phylotypes detected by metabarcoding (Table 2, Appendix A, Appendix A). The metabarcoding runs yielded a total of 193 747 reads from 168 samples (71%) in 69 parks (76%) with an average of 1153 ± 57 reads per site (range: 62–5922). For the sampling sites where *Phytophthora* was detected, there was a mean species richness of 5.8 (in the range from one to 21). *Phytophthora multivora* was the most widely distributed species, detected at 52 parks and 130 samples, followed by *P. arenaria* from 98 samples, *P. amnicola* from 88 samples, *P. cinnamomi* from 81 samples, *P. pseudocryptogea* from 77 samples and *P. nicotianae* from 70 samples (Table 2, Figure 2a). Of the 44 phylotypes, six (*P. capsici*, *P.* sp. pecan, *P. fluvialis*, *P. gonapodyides*, *P.* sp. walnut and *P. fallax*) were rare, accounting for less than 100 of reads and only detected from a single sample (Table 2).

Thirty-six phylotypes were matched with described species. In addition, there were three phylotypes that were matched with ‘designated’ but undescribed species (*P*. sp. pecan, *P*. sp. walnut and *P.* sp. kelmania) and five phylotypes were identified as potential new taxa (*P*. AUS1D, P. AUS2B, P. AUS2B, *P*. AUS8C and P. AUS11A) (Table 2). Fifteen of the species detected (*P*. AUS2B, *P*. AUS2A, *P*. sp. pecan, *P. gonapodyides, P*. sp. walnut, *P. fragariae, P*. AUS8C, *P. drechsleri, P. capensis, P. pachypleura, P. frigida, P*. AUS1D, *P. capsici, P. fallax, P.* AUS11A) have never been isolated in WA (and confirmed by molecular diagnostics), of which the first seven have not been detected in previous metabarcoding studies (Table 2). For 19 of the described species, the first isolates found in WA were from the urban forest (Table 2).

Fifty-five percent of the species found were putatively designated as introduced, while the remainder were designated as putatively native (Table 2, Figure 2). There was unequal phylotype distribution across the *Phytophthora* clades; one third belonged to clade 6, while only two phylotypes were identified from clades 9 and 11 (Table 2, Figure 2b). Species in clades 6, 9 and 11 were predominately considered native, while most of those in clades 2, 7 and 8 were considered introduced (Table 2, Figure 2b). While there were more species detected from clade 6, most of the reads were from clade 2 (Table 2, Figure 2b,c). Two species were detected from clade 9, but they accounted for very few reads (Table 2, Figure 2b,c).

### 3.2. Phytophthora Community

The PERMANOVA analysis of potential relationships of the *Phytophthora* community using presence/absence data revealed a significant (*p* < 0.05) relationship but a small impact (R^2^ < 0.03) for canopy cover and park area (Table 3, Figure 3a). NMDS plots illustrate the complete overlap of *Phytophthora* communities (ellipses overlap) observed for park class and soil family (Figure 3b–c). The other variables show a similar pattern. Twenty-eight of the *Phytophthora* species detected had associations with at least one other species, while 18 species had no associations and were dropped from the matrix (Figure 4a). Species co-occurrence found 89 species pairs had a positive correlation, and three had a negative association (Figure 4a, Appendix A).

### 3.3. Distribution of Phytophthora cinnamomi and P. multivora

*Phytophthora cinnamomi* and *P. multivora* are the dominant invasive species with known impact in Perth, Western Australia. Across the 168 sites, *P. multivora* was detected from 77% of sites and *P. cinnamomi* from 48% of sites. *Phytophthora cinnamomi* was mostly detected at sites also containing *P. multivora*, while *P. multivora* was often detected alone (Figure 4b). *Phytophthora multivora* and *P. cinnamomi* were more likely than chance to occur with one another (Figure 4a), and the two species had the third highest probability of occurrence (37.3%) of all species pairs. Both species were likely to co-occur with the other commonly detected species *P. arenaria, P. capensis, P. thermophila, P. nicotianae and P. citrophthora,* while *P. multivora* was also likely to co-occur with *P. pseudocryptogea, P. inundata, P. constricta and P. litoralis* and *P. cinnamomi* was more likely to co-occur with *P. cambivora* and *P. frigida* (Figure 4a).

## 4. Discussion

Urban parks, whether conservation areas and those that are mostly turf, harbor an unexpected diversity of *Phytophthora.* A decrease in canopy cover and an increase in park area proved to be a weak predictor of the *Phytophthora* community in the urban parks, while canopy health and park class had no influence. Soil baiting recovered only four species, from 10% of sampling sites and 20% of the parks, while metabarcoding of the same samples detected 44 species from 71% of sampling sites and 76% of the parks. These results are comparable to previous studies where both isolations and metabarcoding have been conducted (see Khaliq, Hardy, White and Burgess [16] for a detailed summary). Twenty-five species (including four putative new species and three informally named species) were detected for the first time associated with the urban landscape of Perth, WA. 

### 4.1. Bridgehead Effect

Globalization has increased the frequency of inadvertent introductions of plant pathogens [33], and urban environments provide the opportunity for early detections of these pathogens, as has been demonstrated for *Phytophthora* [13]. Urban forests contain relatively high levels of diversity, and often a mix of native and exotic tree species which provide a range of potential niches for pests and pathogens [4]. In this way, the urban environment functions as a ‘bridgehead’ [34], as points of entry for new pests and pathogens, providing a suitable environment for them to establish and adapt before moving into native ecosystems [7]. In the unified framework for biological invasions, the urban forests play a dual role as both a point of introduction and as an environment which could facilitate establishment [35]. 

Many of the *Phytophthora* species identified in this metabarcoding study were new records for WA. These novel detections were expected, as new species have been detected in all surveys that have used metabarcoding for the detection of *Phytophthora* [14,17,18,19,36]. However, the south-west of WA has had intensive sampling for *Phytophthora* undertaken both in native vegetation, urban parks and gardens for over 40 years [37]. This region has been well explored and in the last decade 20 new *Phytophthora* species have been described [38,39,40,41,42,43,44,45,46,47,48,49]. Hence, unlike many other regions, the baseline of *Phytophthora* species in the south-west of WA is well established. Thus, the detection of 25 species (55% of all species) for the first time in association with the urban environment was unexpected. Of these 15 have never been isolated in WA, which points to the urban environment being a reservoir of potential invasive pathogens. 

### 4.2. Phytophthora Species Detected

Before this study, *P. cinnamomi* and 18 of the other species had previously been isolated from the urban forest of WA. Of these, most were first isolated in the last decade (Barber et al., 2013; Scott et al., 2013). For example, *P. citrophthora* (28 sampling sites) was first recorded in agriculture in 1923 [50] and in the urban forest in 2015 (CPSM diagnostics). This citrus pathogen has been associated with nursery plants imported into WA [50]. *Phytophthora cactorum* (two samples) is pathogenic to more than 200 plant species within different plant families [51] and was recorded in agriculture in 2006, and in the urban forest in 2014 (CPSM diagnostics). *Phytophthora litoralis* (10 samples) was recorded in natural ecosystems in 2006 and in the urban forest in 2011 [41], while *P. boodjera* (24 samples) was first isolated from a native environment in 2016, from the urban environment in 2011 and from a native plant nursery in 2012 [47]. Some *Phytophthora* species have been recorded in agriculture, the urban forest and the natural ecosystem. More prolific, *P. thermophila* (69 samples) appears to be native to WA [41], and has been recorded in water ways, the urban forest, agriculture and natural ecosystems [41,52,53]. These observations together illustrate the potential for species to move between environments.

Of the 25 species not previously isolated from the urban environment of Perth, ten have been isolated previously from other WA environments (native ecosystems, forestry, agriculture). Consequently, their detection within parks and conservation reserves within the urban environment was not unexpected. Two pathogens commonly found in agriculture were detected; *P. erythroseptica,* a potato pathogen [51], from two samples, and *P. fragariae,* a strawberry pathogen [54], from eight samples. This spread of agricultural crop pathogens into the urban forest, and also the detection of species strongly associated with native ecosystems, provides evidence of movement through human activity most likely in the nursery trade or via soil attached to machinery or equipment. Conversely, several *Phytophthora* species considered to be native to WA, but not previously isolated from the urban environment were detected. For example, *P. arenaria* was detected from 98 samples and *P. amnicola* was detected from 88 samples. These findings suggest that these native species were already present although not previously isolated, perhaps because they are not causing disease symptoms. 

Of the 15 species only known in WA from metabarcoding, seven species were detected for the first time in this study. Most of these species are considered to be introduced which supports the role of the urban environment as the pathway by which new *Phytophthora* species eventually reach natural ecosystems and other environments. There is strong evidence from several studies that the first entry into a new country is by distribution via the global nursery trade [7,13,55,56]. The bridgehead effect scenario considers urban environments to be a central source for the spread of *Phytophthora* species into natural ecosystems through many pathways including plant material, aquatic sources, human activities, and equipment. It should be noted that Australia now has very strict quarantine regulations around the movement of plants-for-planting [57], and new detections may be a reflection of an historical rather than a contemporary introduction.

### 4.3. Impact of Parks Class and Area, Canopy Cover and Health on Phytophthora Community

It was assumed that park class would have a significant influence on the *Phytophthora* community as recreational parks have a high rate of human traffic and are mainly comprised of turf, contain many exotic tree species and are watered and fertilized. While conservation areas more closely resemble native forests; they have no cleared areas and receive little management and have less human traffic. It was hypothesized that the managed parks would have more introduced *Phytophthora* species while the conservation parks would have more native *Phytophthora* species. However, the observed *Phytophthora* community was not correlated with any of the measured variables: park class, park area, canopy cover, canopy health or soil type. We suspect that the main reason for the similarity in the *Phytophthora* community was due to the uniformity of the sampling unit which was trees and shrubs native to the region, never turf. Thus, while the majority of the managed parks may be predominantly turf, the areas sampled contained trees. It is also important to note that before the installation of turf the majority of these areas were occupied by native vegetation, some less than 20 years ago. It was also expected that canopy cover and health, as measured by remote sensing, may be a determinant of *Phytophthora* disease, with the lower canopy cover and less healthy canopies being indicative of specific *Phytophthora* communities. However, field validation highlighted that there were other causes for poor tree health, including drought, insect damage and other pathogens [58].

### 4.4. Relationship between Commonly Detected Species

In the current study, the species that were most widely distributed in the urban forests were *P. multivora* (clade 2)*, P. arenaria* (clade 4)*, P. amnicola* (clade 6)*, P. cinnamomi* (clade 7), *P. pseudocryptogea* (clade 8) and *P. nicotianae* (clade 1). All these species are widespread throughout Australia [59]. *Phytophthora arenaria* was described from WA and has never been found outside Australia [42]. *Phytophthora amnicola* was also described from WA and is from a cluster of closely related species commonly found in the south-west of WA [43,60]. *Phytophthora multivora* was described from WA, including isolates from Perth [38], but it has subsequently been reported globally, including from asymptomatic native vegetation in South Africa [61]. The description of *P. pseudocryptogea* included isolates from WA [46]; however it has a comprehensive global distribution in nurseries, orchards and natural forests, and many isolates formerly described as *P. cryptogea* are likely to be reclassified to *P. pseudocryptogea. Phytophthora nicotianae* has a global distribution and causes many well-known diseases, especially in nurseries and agriculture [62]. *Phytophthora cinnamomi* is listed as one of the 100 worst invasive alien species [63] and considered one of the pathogens with the largest global distribution and impact across a range of environments [64]. *Phytophthora arenaria* and *P. amnicola* are considered native to Australia, while the rest are considered to be introduced species and are all well-known pathogens. In the co-occurrence matrix, many of the commonly detected species were positively correlated. Interestingly, these species are very similar to the positively co-occurring species in the Australia-wide metabarcoding survey [59].

Of the commonly detected species *P. cinnamomi* and *P. multivora* are the most widespread and devastating pathogens in natural ecosystems in WA. *Phytophthora cinnamomi* has been present in south-western WA from at least the early 1900′s and is now widely distributed in the region, including Perth. However, in recent years, *P. multivora* has been the most commonly isolated species from dead and dying woody plants in the urban environment in Perth [52], and can infect a wide range of native and introduced host plants common in the urban forest of Perth [65]. In the current study, *P. multivora* dominated the isolations by baiting and there were no recoveries of *P. cinnamomi.* Similarly, *P. multivora* was detected more often in the metabarcoding study (130 samples from 52 parks) than *P. cinnamomi* (81 samples in 40 parks) and, overall, there were five times more reads for *P. multivora.* The study shows that while only *P. multivora* was isolated, *P. cinnamomi* was still present, but probably at a lower inoculum load. In the co-occurrence matrix there were other *Phytophthora* species positively correlated to both *P. multivora* and *P. cinnamomi*, but there were some species more likely to occur with only one of them. This could suggest some shaping of the community, but the main driver of this was not determined in the current study. *Phytophthora cinnamomi* is believed to be a poor saprophyte in soil [66] without long-term survival propagules [67], while *P. multivora* readily produces thick-walled oospores which can be long-term survival propagules [38]. *Phytophthora multivora* may displace *P. cinnamomi* in the future. There was a weak association between soil type and the *Phytophthora* community, with *P. cinnamomi* less common on the more calcareous soil types. *Phytophthora multivora* is known to have a broader pH tolerance [38], and the other *Phytophthora* species found in association with it may have the same trait, but this is yet to be tested.

## 5. Conclusions

Paap, Burgess and Wingfield [7] proposed the dual role of the urban forest in pest invasion biology: firstly, as a niche that allows newly introduced species to adapt to the new environment, and secondly (and concurrently) as a monitoring opportunity to perform targeted searches for newly invasive species before they move into natural ecosystems. This study, while not fulfilling the primary objective to find a correlation between the *Phytophthora* community and tree health, has provided a baseline of *Phytophthora* species in the urban environment of Perth, which includes many species not previously detected in the region. Future studies will be based around understanding which *Phytophthora* species become invasive and the pathways by which they move to the natural ecosystem. Similar studies conducted in Sweden have shown that the urban environment was a reservoir of invasive *Phytophthora* species, where a reduction in species was found as sampling moved to less disturbed sites and then finally into the native forest due to environmental filtering [68]. Future studies will be based around understanding which *Phytophthora* species become invasive and the pathways by which they move into the natural ecosystem.

## Figures and Tables

**Figure 1 microorganisms-08-00973-f001:**
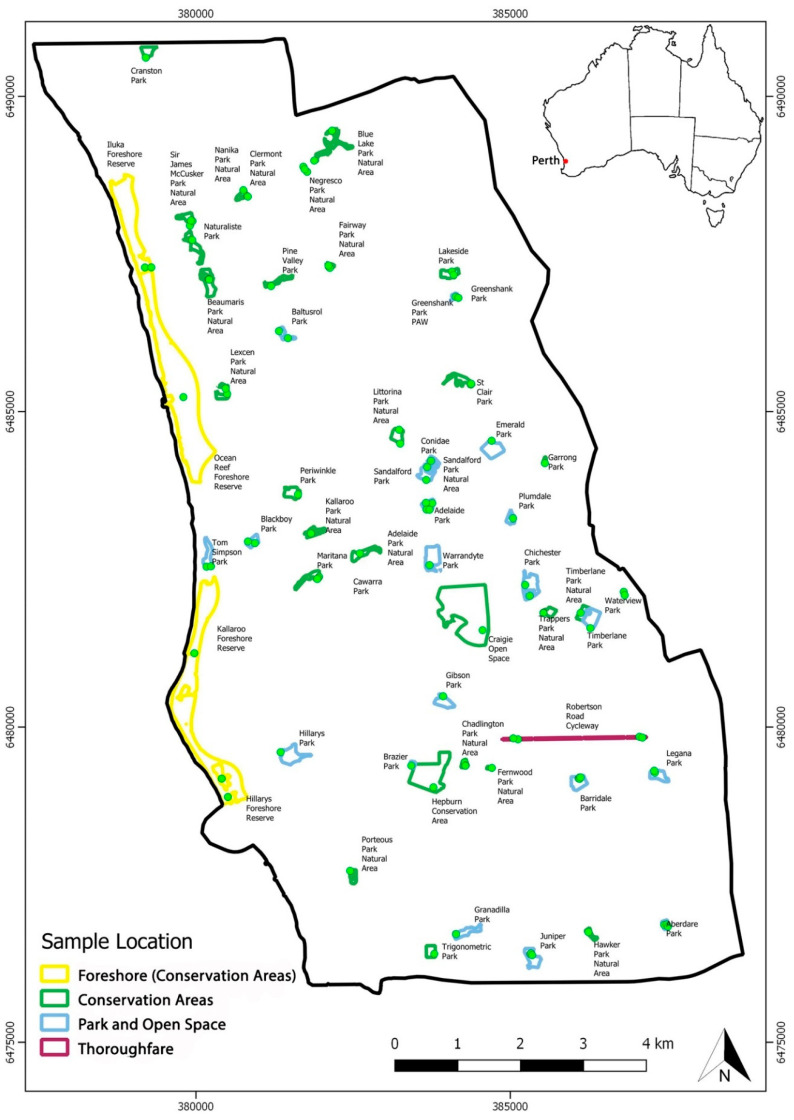
Location of the 91 parks in the City of Joondalup, Western Australia, assessed by remote sensing. Overall, 236 discrete sampling sites (green dots) were selected among these parks in areas where declining vegetation was observed, and soil and roots were sampled for baiting and metabarcoding. A map of Australia is inserted into the top right corner showing the location of Perth.

**Figure 2 microorganisms-08-00973-f002:**
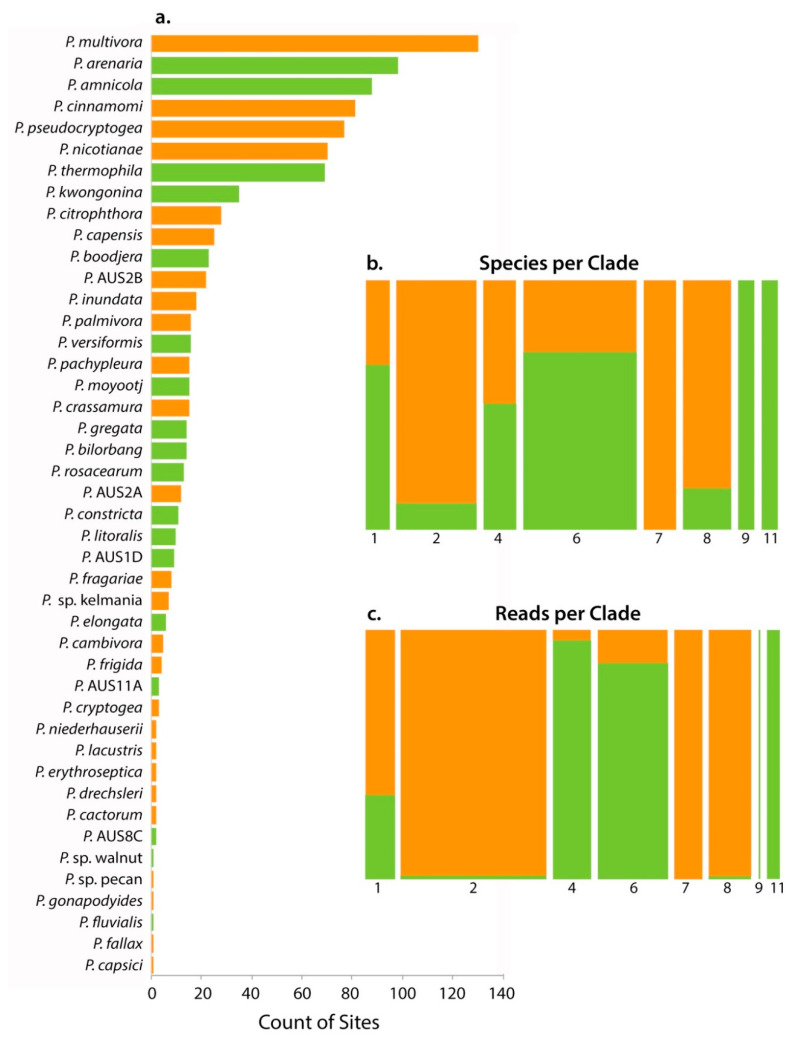
(**a**) Rank-frequency curve for the 44 *Phytophthora* species detected in this study by metabarcoding. Orange bars are the putatively introduced taxa while green denotes putatively native status. Mosaic plot showing the distribution of native and introduced *Phytophthora* species by (**b**) the number of species per clade—the bar width is proportional to the number of species, (**c**) rarefied read number per clade—the bar width is proportional to the number of reads. Bar heights (**b**,**c**) show the relative proportion of native: introduced *Phytophthora* phylotypes per clade.

**Figure 3 microorganisms-08-00973-f003:**
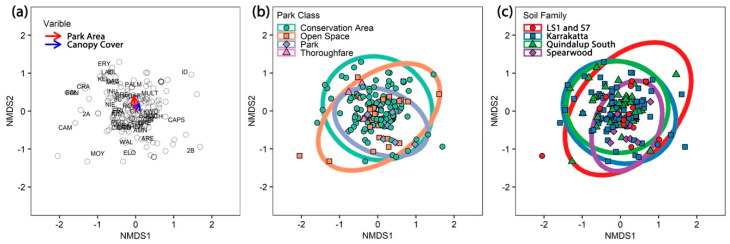
(**a**) Non-metric multidimensional scaling (NMDS) plot showing the significant continuous variables park area and canopy cover. *Phytophthora* species codes and site (hollow points) are displayed on the plot. The arrow length is proportional to the degree of correlations between the variable and the ordination. The arrows are very short indicating minimal correlation. (**b**,**c**) NMDS plots of *Phytophthora* communities showing the complete overlap of ellipses for (**b**) park class, and (**c**) soil family.

**Figure 4 microorganisms-08-00973-f004:**
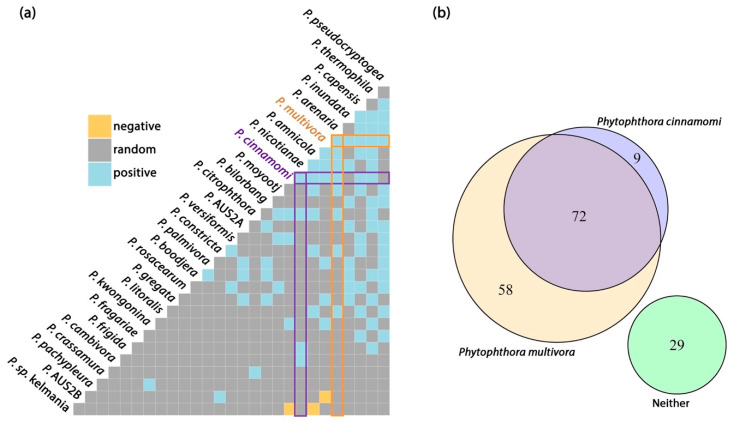
(**a**) Pairwise co-occurrence matrix for a subset of *Phytophthora* species that exceeded a minimum threshold of expected co-occurrence (>1 site). Orange and blue tiles correspond to species pairs that were less or more likey to co-occur than predicted by a null model. (**b**) Venn diagram of detections from 168 sampling sites showing the relationship between the two main invasive species, *Phytophthora cinnamomi* and *P. multivora*.

**Table 1 microorganisms-08-00973-t001:** Data sources and spatial scales for each of the variables examined for sampling sites from which *Phytophthora* was recovered (*n* = 168). The number of sample sites in each category are in brackets.

Factor	Source	Description
Park Class	City of Joondalup	GIS layer units; open space—mainly sport parks (25), conservation areas (117), parks (32) and thoroughfares (4). See Figure 1.
Soil Family	Department of Primary Industries and Regional Development	DAFWA-033 Soil Landscape Mapping WA—best available; LS1 and S7 (24), Quindalup (46), Karrakatta (88) and Spearwood (10) soils.
Park Area	City of Joondalup	Internal GIS layers. Area calculated from polygon layers. Ranged in area from 0.5 to 108 ha. Large park area was deemed to be those sites with area >0.5 SD above the mean, medium ≤0.5 SD from the mean, and small at >0.5 SD below the mean.
Canopy cover	digital multispectral imagery	Standard deviation (SD) of plant cell density (PCD = IR/Red) from pixels within 5 m of the sample point, ranged from 0.005 to 170 ha. High canopy cover was deemed to be those sites with canopy cover >0.5 SD above the mean, medium ≤0.5 SD from the mean, and small at >0.5 SD below the mean.
Canopy health	digital multispectral imagery	SD of Red Edge Extrema Index (REEI = NIR/Red) from pixels of the sample point. The images were acquired in 2012 and 2015, ranging from −17.4 to + 19.5. A -ve value represents a decrease in canopy health.

**Table 2 microorganisms-08-00973-t002:** *Phytophthora* species (*n* = 44) detected by metabarcoding from the urban forest in the City of Joondalup, Western Australia.

*Phytophthora* Species	Clade	No. of Parks	No. of Samples	Reads	Rarefied Read No.	Baiting ^1^	First Record ^2^	Status ^3^
*P. cactorum*	1	2	2	165	13		2014	I
*P. nicotianae*	1	42	70	8262	550	5 (4)	2004	I
*P.* AUS1D ^4^	1	7	9	7667	287			N
*P.* AUS2A ^4,5^	2	11	12	620	33			I
*P.* AUS2B ^4,5^	2	16	22	8912	303			I
*P. capensis* ^4^	2	14	25	6359	251			I
*P. capsici* ^4^	2	1	1	92	12			I
*P. citrophthora*	2	18	28	3113	169		2015	I
*P. elongata*	2	6	6	105	2			N
*P. frigida* ^4^	2	4	4	43	10			I
*P. multivora*	2	52	130	63,649	3251	15 (11)	1985	I
*P. pachypleura* ^4^	2	13	15	2268	83			I
*P. arenaria*	4	54	98	13,221	783	3 (2)		N
*P. boodjera*	4	19	23	4049	263	1 (1)	2011	N
*P. palmivora*	4	14	16	461	41		2011	I
*P.* sp. pecan ^4,5^	4	1	1	2	0			I
*P. amnicola*	6	46	88	12,384	750			N
*P. bilorbang*	6	11	14	962	81			N
*P. crassamura*	6	11	15	6225	224			I
*P. fluvialis*	6	1	1	68	5			N
*P. gonapodyides* ^4,5^	6	1	1	9	1			I
*P. gregata*	6	11	14	300	31		2015	N
*P. inundata*	6	12	18	405	30		2011	I
*P. kwongonina*	6	25	35	4006	188		2010	N
*P. lacustris*	6	2	2	19	3		1995	I
*P. litoralis*	6	6	10	852	35		2011	N
*P. moyootj*	6	11	15	2602	132			N
*P. rosacearum*	6	9	13	1332	67		2015	N?
*P.* sp. walnut ^4,5^	6	1	1	32	2			N?
*P. thermophila*	6	39	69	6667	432		1995	N
*P. cambivora*	7	5	5	1329	67			I
*P. cinnamomi*	7	40	81	11,623	707		∝1980	I
*P. fragariae* ^4,5^	7	6	8	342	14			I
*P. niederhauserii*	7	2	2	276	20		2012	I
*P. cryptogea*	8	3	3	189	12		2015	I
*P. drechsleri* ^4^	8	2	2	27	0			I
*P. erythroseptica*	8	2	2	19	2			I
*P. pseudocryptogea*	8	39	77	14,329	924		2016	I
*P.* sp. kelmania	8	5	7	1832	232		2016	I
*P.* AUS8C ^4,5^	8	1	2	162	10			N?
*P. constricta*	9	7	11	500	20			N
*P. fallax* ^4^	9	1	1	26	3			N?
*P. versiformis*	11	15	16	6119	298		2011	N
*P.* AUS11A ^4^	11	3	3	2126	65			N
Total				193,747	10,416	24 (18)		

^1^ Number of isolates recovered through baiting (number of sites in brackets). ^2^ First recorded in Perth urban forest. ^3^ Status: I = introduced, N= native or N? = putatively native. ^4^ Never isolated in Western Australia (WA). ^5^ First detection in WA by metabarcoding.

**Table 3 microorganisms-08-00973-t003:** PERMANOVA analysis showing correlation between five variables and the *Phytophthora* community based both on presence/absence data and number of reads. Bold *P* values indicate a significant relationship. Low R^2^ values indicate that while significant these variables had little effect on the *Phytophthora* community.

		Presence	Abundance
Factor	df	SS	MS	F-Value	R^2^	*P*-Value	SS	MS	F-Value	R^2^	*P*-Value
Global											
Park Area	1	1.64	1.64	5.51	0.03	**0.0001**	1.52	1.52	4.71	0.03	**0.0001**
Park Class	3	1.06	0.35	1.19	0.02	0.1809	0.94	0.31	0.97	0.02	0.5009
Canopy Cover	1	0.90	0.90	3.01	0.02	**0.0001**	1.51	1.51	4.68	0.03	**0.0001**
Canopy Health	1	0.35	0.35	1.16	0.01	0.2890	0.40	0.40	1.25	0.01	0.2256
Soil Family	3	1.33	0.44	1.48	0.03	**0.0204**	1.35	0.45	1.39	0.02	0.0532
Residuals	158	47.04	0.30		0.89		51.08			0.89	
Total	167	52.31			1.00		56.81			1.00

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
