# Peer review of "Association of Phytophthora with Declining Vegetation in an Urban Forest Environment"

_microorganisms, 2020, doi:10.3390/microorganisms8070973_

Round 1

Reviewer 1 Report

The manuscript “Association of Phytophthora with declining vegetation in an urban forest environment” by Khdiar & al report a survey of Phytophthora communities in urban tree environments in a town of Western Australia, by both baiting and metabarcoding. The work is well conducted and bring out a lot of interesting and useful information on the potential role of urban forests as proper establishment location for many invasive Phytophthora species. I would strongly recommend it publication in Microorganisms and just have some minor remarks to do.

  • I am not completely sur I properly understood what “site” refers to. Is it one sampled plant/group of plant giving one soil sample? You may want to clarify, as it is often used (in particular in Table 2).
  • In figure 2 / table 2, you are classifying all taxa as putatively invasive / native. You say nothing on how this is done. While it appears “easy” for well known described taxa (P. cinnamomi, P. multivora), I was wondered how this may be performed for putative undescribed phylotype such as P. AUS2A / P. AUS1D for which nothing or just little is known. I feel an explanation is needed in Material and Methods.
  • Table 2 relation rarified read nb / isolation. P. cinnamomi not baited though quite frequent. Normal? Shouldn’t it be quite readily baited?
  • 36/44 match described species – not bad ! Well discussed in 4.2
  • “Park” categories in Fig 3b do not match those in Table 1. Clarify. Arrows in Fig3a are difficult to see. Is it possible to improve?
  • Although the Fig.4 is quite clear, can’t you formally test the P. cinnamomi / P. multivora co-occurrence (with a chi-square test for example, but I do not see clearly whether there is some dependency to take into account)?
  • L251-257. You may want to place your result in the general invasion framework outlined by Blackburn et al (2011) (10.1016/j.tree.2011.03.023).
  • L303-316. No difference between park / natural areas. Nothing is said about planting frequency in these environments. Some information?

Reviewer 2 Report

Paper "Association of Phytophthora with declining vegetation in an urban forest environment” by Khdiar et al., gives an insight into Phytophthora spp. presence and distribution in one part of Australia (Western Australia), and investigates links among different factors (such as soil type, tree health status, …) and the incidence of Phytophthora spp.

I only have some minor comments:

Line 19 (and throughout the text): I would suggest that rather than “sites”, different phrase is used (eg. “sampling site”; “sampling location”)

L 22-23: Reading the sentence about baiting and metabarcoding it can be misunderstood, that first the roots were baited and then DNA was extracted. I would suggest to rephrase the sentence to be more clear.

L 30-31: contradiction? “While park area and canopy cover had a significant effect on Phytophthora community the R2 values were very low indicating they have had little effect.”

L 60-71: this paragraph about remote sensing in the introduction is a bit out of the introduction story. I would delete it, or make it shorter and transfer it to discussion part.

L 83-84: could some additional info be added what is meant under “natural areas” and “turf”?

Fig 1: I would suggest to implement a small map of Australia, where the studied area would be marked.

FIG 1: Add the legend for “green dots”

L 113-114: how long were the samples stored under room temperatures? Why not in the fridge?

L 143: add the names of the Phytophthora clades. Also, at the link provided the names of the clades cannot be easily find.

L 200: were these species isolated in Australia?

L 204-205, Table 2: how was the designation for each species assign (based on what was the species regarded as Invasive or Native)? Maybe add references to the Table 2 for each species?
